# Revisiting Knowledge Tracing: A Simple and Powerful Model

## ABSTRACT

Advances in multimedia technology and its widespread application in education have made multimedia learning increasingly important. Knowledge Tracing (KT) is the key technology for achieving adaptive multimedia learning, aiming to monitor the degree of knowledge acquisition and predict students' performance during the learning process. Current KT research is dedicated to enhancing the performance of KT problems by integrating the most advanced deep learning techniques. However, this has led to increasingly complex models, which reduce model usability and divert researchers' attention away from exploring the core issues of KT. This paper aims to tackle the fundamental challenges of KT tasks, including the knowledge state representation and the core architecture design, and investigate a novel KT model that is both simple and powerful. We have revisited the KT task and propose the ReKT model. First, taking inspiration from the decision-making process of human teachers, we model the knowledge state of students from three distinct perspectives: questions, concepts, and domains. Second, building upon human cognitive development models, such as constructivism, we have designed a Forget-Response-Update (FRU) framework to serve as the core architecture for the KT task. The FRU is composed of just two linear regression units, making it an extremely lightweight framework. Extensive comparisons were conducted with 22 state-of-the-art KT models on 7 publicly available datasets. The experimental results demonstrate that ReKT outperforms all the comparative methods in question-based KT tasks, and consistently achieves the best (in most cases) or near-best performance in concept-based KT tasks. Furthermore, in comparison to other KT core architectures like Transformers or LSTMs, the FRU achieves superior prediction performance with approximately only 38% computing resources. Through an exploration of the ReKT model that is both simple and powerful, is able to offer new insights to future KT research. The code is in the supplementary materials.

## CCS CONCEPTS

• **Information systems** → **Data mining**; • **Applied computing** → **E-learning**.

## KEYWORDS

adaptive multimedia learning; knowledge tracing; student performance prediction; educational data mining

*ACM MM, 2024, Melbourne, Australia*

© 2024 Copyright held by the owner/author(s). Publication rights licensed to ACM.
ACM ISBN 978-x-xxxx-xxxx-x/YY/MM
https://doi.org/10.1145/nnnnnnn.nnnnnnn

## 1 INTRODUCTION

Given the rapid advancement of multimedia technology in education, multimedia learning has become indispensable in modern teaching methods[41]. This underscores the significance of adaptive multimedia learning, prioritizing the customization of educational experiences. At the core of achieving this goal lies Knowledge Tracing technology. Knowledge Tracing (KT) is the key technology for achieving adaptive multimedia learning, aiming to monitor the degree of knowledge acquisition (knowledge state) and predict students' performance during the learning process. Effectively addressing KT task can assist teachers in gaining insights into students' learning progress, enabling them to tailor teaching strategies and offer personalized guidance [3].

As shown in Figure 1, the example of KT involves each question being associated with one or more concepts. Students practice with different questions, and the aim of KT is to predict the probability of them answering the next question correctly. Often, due to the lack of question information in certain datasets, KT can be further categorized into question-based KT and concept-based KT for uniformity. Additionally, to simplify the KT task, multiple concepts are sometimes amalgamated into a new concept [44]. Many methods [2, 21, 32] have been proposed for KT, which predict students' future performance by tracing their knowledge state. Thus, the key aspect of KT is effectively tracing and representing students' knowledge state. However, most KT methods exclusively focus on modeling either question-based KT or concept-based KT, without exploring their potential in both of these scenarios.

In recent years, with the advancement of deep learning technologies, KT models based on methods such as Transformers [10, 12], graph neural networks [37, 38, 43], and contrastive learning [18, 49] have emerged one after another. Although these works have made significant contributions to in-depth research on KT, we have observed a strange phenomenon: the current mainstream of KT research seems to rely on cutting-edge technologies from other domains to build complex models, while efforts to delve deeper into the KT problem itself appear to have made limited progress. While works like LPKT [32] and LBKT [45] explore the impact of various behaviors during student learning processes, they often depend on specialized and intricate architectures, making it challenging for subsequent research to derive substantial insights from them. We hope to Revisit Knowledge Tracing (ReKT) and design a model that is as simple and powerful as possible.

We start by addressing the two fundamental challenges of the KT task: (1) How to represent a student's knowledge state from learning data; (2) How to design a core architecture that is as simple as possible and suitable for KT. For the first challenge, in fact, long before we began using KT models to solve this problem, teachers had already been engaged in a similar process. When teachers assess a student's ability to solve a specific question, they consider several key factors. First, they assess the student's previous performance on the same question. Second, they consider the student's performance on similar questions previously. Finally, if the student has

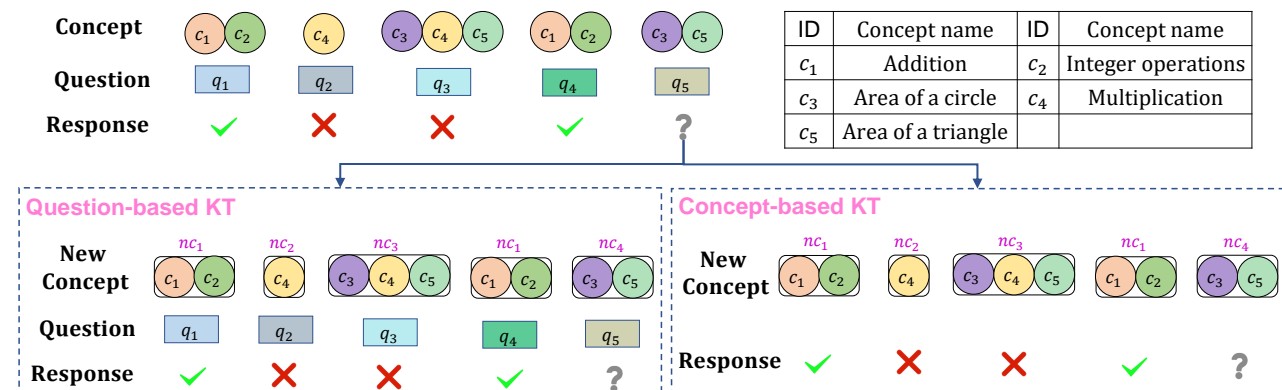

**Figure 1: A simple example of knowledge tracing, "Response" indicates the student's answer.**

not encountered this question or similar ones before, the teacher consider the student's overall historical performance. Inspired by this process, we model student's knowledge state from three distinct perspectives: questions, concepts, and domains. Regarding the second challenge, drawing inspiration from human cognitive development models [4], which emphasizes that changes in human knowledge state (Response) are mainly affected by two main psychological processes: internalization and forgetting. Internalization emphasizes the updating (Update) process of knowledge state based on environmental stimuli, while forgetting (Forget) emphasizes the natural changing process of knowledge state over time. Therefore, we model these core processes and design a lightweight core architecture called FRU (Forget-Response-Update) for KT tasks.

Specifically, we propose ReKT, which is both simple and powerful. It traces student's knowledge state from three distinct perspectives: (1) It traces students' **question** knowledge state from interaction history limited to specific questions, indicating their mastery of specific questions; (2) It traces students' **concept** knowledge state from interaction history limited to specific concepts (where two questions with the same concept are considered similar), indicating their mastery of questions encompassing those specific concepts; (3) It traces students' **domain** knowledge state from their entire interaction history, indicating their mastery of questions spanning the entire domain (i.e., overall performance). We combine these three to comprehensively represent students' knowledge state. Furthermore, we've designed a lightweight core architecture called FRU for KT tasks. It first calculates the forgetting of the student's knowledge state based on the interval time, then responses based on the knowledge state, and finally updates the knowledge state based on the learning interaction. It's worth mentioning that FRU comprises only two linear regression units. Experimental results on 7 publicly available datasets, comparing ReKT with 22 state-of-the-art KT models, including question-based KT and concept-based KT, show that in most cases, ReKT significantly outperforms other models. This demonstrates that even without relying on highly complex models or cutting-edge technologies, by delving deeply into the characteristics of the KT task, it is possible to construct a model that is both simple and powerful. We believe that ReKT has the potential to offer a wealth of new inspiration and insights for future KT research.

## 2 RELATED WORK

KT was proposed by [9] in 1994. Classic knowledge tracing methods can be categorized into bayesian-based methods such as BKT[14, 50] and factor analysis-based methods such as LFA[5], AFM[6], PFA[28], KTM[40]. In recent years, with the advancement of deep learning, an increasing number of methods have been employed to tackle KT tasks. We categorize the existing methods into the following two types based on the nature of the KT task:

**Concept-based KT**: This category of KT methods aims to predict the student's mastery of specific concepts. DKT [29] is regarded as a representative method for concept-based KT, utilizing an LSTM to construct the student's knowledge state. DKVMN [51] employs a dynamic key-value memory network to capture the student's knowledge state. SAKT [26] models the relationship between students and concepts using self-attention mechanisms to derive the knowledge state. SKVMN [1] employs an enhanced LSTM for modeling students' knowledge state and updates it based on students' responses to relevant questions. GKT [24] propagates a student's knowledge state over a graph structure. SKT [38] takes into account various relationships between concepts to simulate the propagation of knowledge states. ATKT [13] utilizes adversarial learning to represent students' knowledge states in a more robust manner. CL4KT [18] designs various data augmentation strategies to enhance the representational capacity of the knowledge state.

**Question-based KT**: Building upon concept-based KT, this category of methods includes additional question information to predict a student's performance on specific questions. AKT [12] is considered a representative method for question-based KT, utilizing a context-aware Transformer architecture to account for forgetting behavior in tracing students' knowledge state. SAINT [8] constructs the student's knowledge state entirely using the Transformer framework. GIKT [46] employs GCN[16] to uncover relationships between questions and concepts, and incorporates an interactive module to capture students' knowledge states. DIMKT [31] extensively mines question difficulty information to model the student's knowledge state. simpleKT [21] simplifies AKT to trace students' knowledge state. DTransformer [49] employs contrastive learning to maintain a stable knowledge state. AT-DKT [20] enhances knowledge state representation by introducing two extra tasks related to question design.

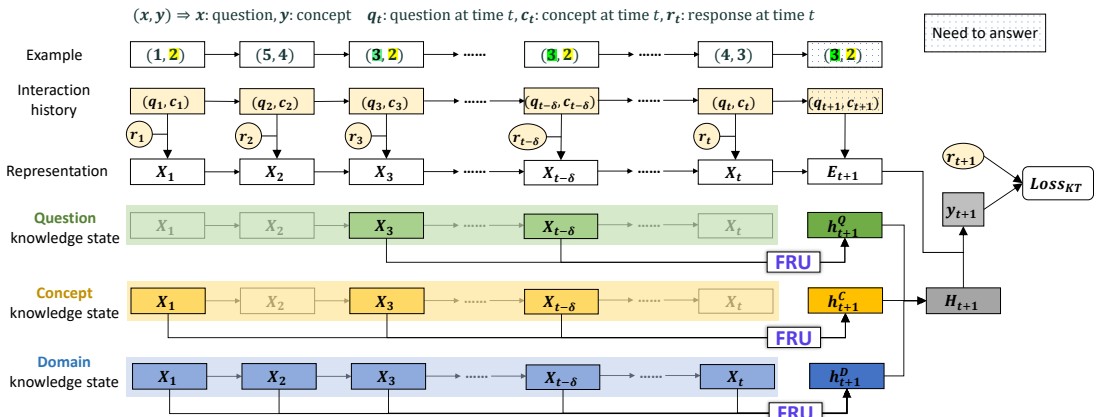

Figure 2: The overall framework of ReKT. ReKT constructs the question and concept knowledge states from the interaction history limited to the current question and concept. Additionally, it constructs the domain knowledge state using the entire interaction history.

## 3 METHOD

Figure 2 showcases ReKT's framework. We'll introduce KT's problem formulation and then detail ReKT's modules.

### 3.1 Problem Formulation

The KT task can be defined as follows: Given a student's interaction history sequence $L = \{(q_1, c_1, r_1), (q_2, c_2, r_2), ..., (q_t, c_t, r_t)\}$, where $q_t$ is the question at time $t$, $c_t$ signifies the concept associated with question $q_t$, and $r_t \in \{0, 1\}$ shows if the student's response to $q_t$ is correct. KT's aim is to predict the probability of the student answering the next question $q_{t+1}$ correctly. For concept-based KT, which lacks specific question data, concepts are treated as equivalent representations of questions[23, 51]. So, the goal is to predict the probability of the student answering the next concept $c_{t+1}$ correctly. Please note that if a question involves multiple concepts, we combine them into a new concept[44].

### 3.2 Representation of questions, concepts and responses

Effectively representing questions, concepts, and responses is important for KT, and establishing their relationships forms a highly effective approach. Let's denote the feature matrix as follows: $Q \in \mathbb{R}^{n \times d}$ for questions, $C \in \mathbb{R}^{m \times d}$ for concepts, and $R \in \mathbb{R}^{2 \times d}$ for responses. Here, $n$ represents the total number of questions, $m$ is the total number of concepts, and $d$ signifies the feature dimension. Inspired by the classic psychological measurement theory of Rasch [30], which explicitly employs scalars to represent question difficulty. For the current question $q_{t+1}$ and its associated concept $c_{t+1}$, their feature representation $E_{t+1} = Q_{q_{t+1}} + C_{c_{t+1}} + diff_{q_{t+1}} * V_{c_{t+1}}$. Here, $Q_{q_{t+1}}$ denotes the $q_{t+1}$-th row of $Q$, $C_{c_{t+1}}$ corresponds to the $c_{t+1}$-th row of $C$, $diff_{q_{t+1}}$ represents the difficulty of $q_{t+1}$ and is a scalar, and $V_{c_{t+1}} \in \mathbb{R}^{1 \times d}$ captures the extent of variation of the question with respect to its concept. For any given moment $t$ within the interaction history $L$, where $L_t = (q_t, c_t, r_t)$, the feature representation $X_t$ for $L_t$ is calculated as $X_t = E_t + R_{r_t}$, where $R_{r_t}$ corresponds to the $r_t$-th row of $R$.

### 3.3 Representation of knowledge state

The representation of a student's knowledge state is the most fundamental issue in KT, and ReKT constructs three types of knowledge states for a student from the interaction history $L = (q_1, c_1, r_1), (q_2, c_2, r_2), ..., (q_t, c_t, r_t)$: question, concept, and domain knowledge state. For the current question $q_{t+1}$ and its corresponding concept $c_{t+1}$:

For the aspect of question knowledge state: We construct it from the interaction history $L_{t+1}^Q$ limited to the current question $q_{t+1}$. Here, $L_{t+1}^Q = \bigcup_j L_j$, if $q_j == q_{t+1}, j < t + 1$. $\bigcup$ represents the union operation. The feature representation of $L_{t+1}^Q$ is denoted as $X_{t+1}^Q$, as shown in Figure 2, where $X_{t+1}^Q = \{X_3, X_{t-\delta}\}$. As shown in the example, they all include the current question 3. Thus, the question knowledge state $h_{t+1}^Q = $ FRU $(X_{t+1}^Q \cup X_{t+1})$. The FRU architecture is shown in Figure 3, and we'll provide a detailed explanation of it in the next section. Note the inclusion of an added term, $X_{t+1}$, here. This serves the purpose of informing the FRU about the current time, i.e., $t + 1$.

For the aspect of concept knowledge state: We construct it from the interaction history $L_{t+1}^C$ limited to the current concept $c_{t+1}$. Here, $L_{t+1}^C = \bigcup_j L_j$, if $c_j == c_{t+1}, j < t + 1$. The feature representation of $L_{t+1}^C$ is denoted as $X_{t+1}^C$, as shown in Figure 2, where $X_{t+1}^C = \{X_1, X_3, X_{t-\delta}\}$. As shown in the example, they all include the current concept 2. Therefore, the concept knowledge state $h_{t+1}^C = $ FRU $(X_{t+1}^C \cup X_{t+1})$. The inclusion of $X_{t+1}$ here also aims to inform the FRU about the current moment. The FRU here is distinct from the earlier mentioned FRU parameter.

For the aspect of domain knowledge state: We construct it from the entire interaction history $L$. The feature representation of $L$ is denoted as $X$, as shown in Figure 2, where $X = \{X_1, X_2, X_3, ... X_{t-\delta}, ..., X_t\}$. Therefore, the domain knowledge state $h_{t+1}^D = $ FRU $(X \cup X_{t+1})$. Consistent with the above, here $X_{t+1}$ is used to inform the FRU of the current moment, and the FRU parameters here are independent of the previous FRU's parameters.

## 3.4 FRU framework

Inspired by human cognitive development models [4], we design a lightweight core architecture called FRU (Forget-Response-Update) for KT tasks, as shown in Figure 3. It first takes into account the process of students forgetting their knowledge state, then responses to specific questions based on knowledge state, and ultimately updates the knowledge state based on response accuracy. Specifically, let the current moment be $t$, the last relevant moment is $t - \alpha$ (this allows FRU to handle time series with varying intervals), $Z_{t-\alpha} \in \mathbb{R}^{1 \times d}$ represents the knowledge state at the time $t - \alpha$. First, we remember that the interval time $\alpha$ is represented by features as $I_\alpha \in \mathbb{R}^{1 \times d}$, then at the current moment $t$, the degree of forgetting of $Z_{t-\alpha}$ can be calculated as $f_t = Sigmoid([Z_{t-\alpha} \oplus I_\alpha]W_1 + b_1)$, where $W_1$ and $b_1$ are all learnable parameters. Then the knowledge state of the current response $Response_t = f_t * Z_{t-\alpha}$. Finally, the student will update the knowledge state according to the feature representation $X_t$ of current interaction, then the student's current knowledge state $Z_t = Response_t + Tanh([Response_t \oplus X_t]W_2 + b_2)$, Among them, $W_2$ and $b_2$ are all learnable parameters. The subsequent processing flow can be deduced by analogy.

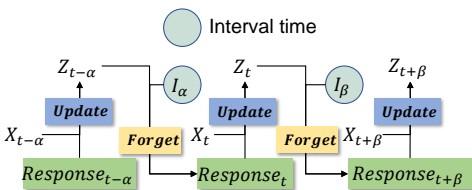

**Figure 3: FRU framework.**

We record the interaction sequence input by FRU as $X_t^\xi = \{..., X_{t-\alpha}, X_t\}$, then FRU can be abstracted as: $Response_t = $ FRU $(X_t^\xi)$. Clearly, the FRU framework is very lightweight, as it consists of only two linear regression units. However, it aligns remarkably well with students' cognitive learning processes, and subsequent experiments will demonstrate its effectiveness.

## 3.5 Prediction and training

For the question $E_{t+1}$ that needs to be answered after feature representation and the students' three knowledge states $h_{t+1}^Q$, $h_{t+1}^C$ and $h_{t+1}^D$, then the student's current final knowledge state $H_{t+1} = h_{t+1}^Q \oplus h_{t+1}^C \oplus h_{t+1}^D$. It's worth noting that here we use concatenation rather than attention mechanisms to combine them, as we consider that different knowledge states play distinct roles. Finally, based on the student's current knowledge state $H_{t+1}$, we predict the student's response $y_{t+1}$ for answering $E_{t+1}$, in other words, $y_{t+1} = Sigmoid(W_4ReLU(W_3[E_{t+1} \oplus H_{t+1}] + b_3) + b_4)$, where $W_3$, $b_3$, $W_4$ and $b_4$ are all learnable parameters. The knowledge tracing loss $Loss_{KT}$ is computed as follows:

$$Loss_{KT} = -\sum_{t=1}^{T-1}(r_{t+1}log y_{t+1} + (1 - r_{t+1})log(1 - y_{t+1}))$$

Here, $T$ represents the total number of time steps, and $r_{t+1}$ denotes the actual response result from the student at time step $t + 1$. We use Adam[15] to optimize model parameters.

## 4 EXPERIMENTS

### 4.1 Datasets

We evaluated the performance of ReKT on 7 publicly available commonly used datasets: ASSIST09[1], ASSIST12[2], ASSIST15[3], ASSIST17[4], Statics2011[5], EdNet[6], and Eedi[7]. The statistical information for these datasets is provided in supplementary materials. Based on previous research, for the ASSIST series datasets, we removed scaffold questions and records without concepts[12]. Additionally, multiple concepts were merged into new concepts[44]. For the ASSIST15 dataset, records with an "isCorrect" field not equal to 0 or 1 were removed[1]. In the case of Statics2011, original question indexes and step indexes were combined into a new question index. If the same question was answered consecutively, only the first answer was retained. Moreover, due to the large scale of ASSIST12, EdNet, and Eedi datasets, and limitations in computational resources, we randomly sampled records from 5000 students[19, 46].

### 4.2 Baseline

To evaluate ReKT's performance, we compared it to baseline models of concept-based KT and question-based KT.

*4.2.1 Concept-based KT baseline.* **BKT**[9]: The first knowledge tracing method employs hidden markov models to trace students' knowledge states. **DKT**[29]: The first deep learning-based knowledge tracing model that employs LSTM to trace students' knowledge states. **DKVMN**[51]: Utilizes a dynamic key-value memory network to model students' knowledge states. **DKT+**[48]: Enhances DKT by addressing inconsistent knowledge states. **KQN**[17]: Predicts students' performance using knowledge states and concept encoders. **DeepIRT**[47]: Introduces IRT[42] to DKVMN for improved interpretability of predictions. **DKT+forgetting**[23]: Enhancing DKT by incorporating students' forgetting behaviors. **SAKT**[26]: The first self-attention mechanism-based knowledge tracing model that attempts to capture the relationship between students and concepts. **GKT**[24]: The first graph-based knowledge tracing model that propagates students' knowledge states within the proposed graph. **AKT-concept**[12]: A variant of AKT where inputs are limited to concepts. **SAINT-concept**[8]: A variant of SAINT where inputs are limited to concepts. **ATKT**[13]: Maintaining stable knowledge states through adversarial learning. **CL4KT**[18]: Introduces multiple data augmentation strategies and employs contrastive learning to alleviate sparsity in student-interaction data.

*4.2.2 Question-based KT baseline.* **KTM**[40]: Modeling students' knowledge states using Factorization Machines. **AKT**[12]: Proposes a context-aware Transformer architecture to simulate students' forgetting behavior for knowledge state tracing. **SAINT**[8]: Fully employs a Transformer [39] architecture to model students' knowledge states. **PEBG+DKT**[19]: Enhances DKT by deeply exploring the

---

[1]https://sites.google.com/site/assistmentsdata/home/2009-2010-assistment-data
[2]https://sites.google.com/site/assistmentsdata/home/2012-13-school-data-with-affect
[3]https://sites.google.com/site/assistmentsdata/home/2015-assistments-skill-builder-data
[4]https://sites.google.com/view/assistmentsdatamining/dataset
[5]https://pslcdatashop.web.cmu.edu/DatasetInfo?datasetId=507
[6]https://github.com/riiid/ednet
[7]https://eedi.com/projects/neurips-education-challenge

**Table 1: Comparison of ReKT-concept and concept-based KT baseline on 7 datasets. Best results in bold, next best underlined.**

| Model | ASSIST09 | | ASSIST12 | | ASSIST15 | | ASSIST17 | | Statics2011 | | EdNet | | Eedi | |
|---|---|---|---|---|---|---|---|---|---|---|---|---|---|---|
| | AUC | ACC | AUC | ACC | AUC | ACC | AUC | ACC | AUC | ACC | AUC | ACC | AUC | ACC |
| BKT | 0.7180 | 0.6922 | 0.6613 | 0.7096 | 0.6739 | 0.7423 | 0.6519 | 0.6290 | 0.7444 | 0.7973 | 0.6737 | 0.7018 | 0.6828 | 0.6680 |
| DKT | 0.7684 | 0.7297 | 0.7328 | 0.7367 | 0.7323 | 0.7536 | 0.7188 | 0.6673 | 0.8483 | 0.8275 | 0.7006 | 0.7129 | 0.7629 | 0.7182 |
| DKVMN | 0.7629 | 0.7266 | 0.7228 | 0.7329 | 0.7310 | 0.7540 | 0.7142 | 0.6639 | 0.8363 | 0.8224 | 0.6975 | 0.7120 | 0.7590 | 0.7162 |
| DKT+ | **0.7783** | 0.7337 | 0.7373 | 0.7350 | 0.7304 | 0.7542 | 0.7095 | 0.6622 | **0.8718** | 0.8270 | 0.7028 | 0.6698 | 0.7484 | 0.7079 |
| KQN | 0.7546 | 0.7249 | 0.7230 | 0.7330 | 0.7263 | 0.7544 | 0.7065 | 0.6611 | 0.8031 | 0.7969 | 0.6909 | 0.7117 | 0.7583 | 0.7143 |
| DeepIRT | 0.7657 | 0.7279 | 0.7253 | 0.7345 | 0.7283 | 0.7530 | 0.7174 | 0.6647 | 0.8408 | 0.8262 | 0.6997 | 0.7124 | 0.7609 | 0.7173 |
| DKT+forgetting | 0.7717 | 0.7295 | 0.7362 | 0.7359 | 0.7529 | 0.7607 | 0.7165 | 0.6665 | 0.8467 | 0.8182 | 0.7018 | **0.7159** | 0.7642 | 0.7186 |
| SAKT | 0.7564 | 0.7192 | 0.7296 | 0.7348 | 0.7436 | 0.7558 | 0.7079 | 0.6596 | 0.8092 | 0.8111 | 0.6956 | 0.7115 | 0.7556 | 0.7123 |
| GKT | 0.7666 | 0.7290 | 0.7261 | 0.7333 | 0.7289 | 0.7528 | 0.7203 | 0.6685 | 0.8384 | 0.8224 | 0.6943 | 0.7104 | 0.7618 | 0.7170 |
| AKT-concept | 0.7668 | 0.7280 | **0.7384** | 0.7390 | 0.7312 | 0.7557 | 0.7157 | 0.6637 | 0.8384 | 0.8204 | 0.6987 | 0.7111 | 0.7626 | 0.7166 |
| SAINT-concept | 0.7487 | 0.7140 | 0.7289 | 0.7353 | 0.7365 | 0.7572 | 0.7013 | 0.6560 | 0.7598 | 0.7940 | 0.6974 | 0.7096 | 0.7589 | 0.7130 |
| ATKT | 0.7735 | 0.7332 | 0.7347 | 0.7363 | 0.7311 | 0.7555 | 0.7198 | 0.6699 | 0.8175 | 0.7991 | 0.7027 | 0.7109 | 0.7663 | 0.7195 |
| CL4KT | 0.7626 | 0.7275 | 0.7236 | 0.7331 | 0.7310 | 0.7549 | 0.7139 | 0.6615 | 0.8251 | 0.8170 | 0.6965 | 0.7118 | 0.7583 | 0.7147 |
| ReKT-concept | 0.7737 | **0.7340** | 0.7359 | 0.7385 | **0.7531** | **0.7624** | **0.7237** | 0.6690 | 0.8546 | **0.8319** | 0.7096 | 0.7153 | **0.7693** | **0.7208** |

**Table 2: Comparison of ReKT and question-based KT baseline on 6 datasets. Best results in bold, next best underlined.**

| Model | ASSIST09 | | ASSIST12 | | ASSIST17 | | Statics2011 | | EdNet | | Eedi | |
|---|---|---|---|---|---|---|---|---|---|---|---|---|
| | AUC | ACC | AUC | ACC | AUC | ACC | AUC | ACC | AUC | ACC | AUC | ACC |
| KTM | 0.7190 | 0.7020 | 0.7093 | 0.7245 | 0.7248 | 0.6686 | 0.8157 | 0.8071 | 0.7583 | 0.7293 | 0.7035 | 0.6792 |
| AKT | 0.7850 | 0.7429 | 0.7830 | 0.7599 | 0.7572 | 0.6916 | 0.8718 | 0.8376 | 0.7616 | 0.7372 | 0.7882 | 0.7340 |
| SAINT | 0.7515 | 0.7134 | 0.7643 | 0.7477 | 0.7537 | 0.6894 | 0.8279 | 0.8110 | 0.7621 | 0.7370 | 0.7866 | 0.7293 |
| PEBG+DKT | 0.7738 | 0.7329 | 0.7518 | 0.7495 | 0.7619 | 0.6949 | 0.8655 | 0.8368 | 0.7571 | 0.7366 | 0.7853 | 0.7310 |
| GIKT | 0.7726 | 0.7301 | 0.7672 | 0.7506 | 0.7723 | 0.6989 | 0.8834 | 0.8428 | 0.7640 | 0.7366 | 0.7924 | 0.7362 |
| CDKT | 0.7733 | 0.7297 | 0.7720 | 0.7547 | 0.7709 | 0.7019 | 0.8872 | 0.8510 | 0.7645 | 0.7386 | 0.7920 | 0.7360 |
| DIMKT | 0.7704 | 0.7310 | 0.7621 | 0.7484 | 0.7682 | 0.6993 | 0.8897 | 0.8501 | 0.7623 | 0.7368 | 0.7908 | 0.7338 |
| QIKT | 0.7801 | 0.7377 | 0.7707 | 0.7529 | 0.7645 | 0.6985 | 0.8817 | 0.8482 | 0.7579 | 0.7327 | 0.7932 | 0.7363 |
| simpleKT | 0.7772 | 0.7315 | 0.7786 | 0.7571 | 0.7570 | 0.6899 | 0.8614 | 0.8350 | 0.7627 | 0.7373 | 0.7885 | 0.7307 |
| AT-DKT | 0.7671 | 0.7293 | 0.7425 | 0.7405 | 0.7265 | 0.6702 | 0.8687 | 0.8386 | 0.7039 | 0.7136 | 0.7649 | 0.7180 |
| DTransformer | 0.7646 | 0.7223 | 0.7672 | 0.7515 | 0.7538 | 0.6898 | 0.8686 | 0.8513 | 0.7501 | 0.6954 | 0.7531 | 0.7315 |
| ReKT | **0.7917** | **0.7449** | **0.7852** | **0.7609** | **0.7814** | **0.7102** | **0.8967** | **0.8568** | **0.7752** | **0.7447** | **0.7971** | **0.7397** |

relationship between questions and concepts to obtain pre-trained question representations. **GIKT**[46]: Utilizes GCN[16] to aggregate relationships between questions and concepts. **CDKT**[11]: Enhances DKT by employing contrastive learning to learn informative question representations. **DIMKT**[31]: Explores question difficulty information and its relationship with students' knowledge states. **QIKT**[7]: Designs question-sensitive cognitive representations for modeling student knowledge states. **simpleKT**[21]: A simplified version of AKT, which simplifies the architecture of AKT without sacrificing too much performance. **AT-DKT**[20]: Enhances DKT by introducing two additional question-related tasks. **DTransformer**[49]: Proposing a novel architecture to trace students' patterns of learning activities

## 4.3 Experimental setting

We implemented ReKT using PyTorch[27]. For each dataset, we split 80% of all the sequences as the training set, 20% as the test set[46, 49]. The learning rate was set to 0.002, batch size to 80, and feature dimension $d$ to 128. Additionally, we applied L2 regularization to the model weights with a decay coefficient of 1e-5. Sequences with a length of less than 3 were removed. To handle variable sequence lengths, all sequences were padded to a consistent length of 200. To mitigate overfitting, we introduced a dropout rate of 0.4. ReKT was trained on a Linux server with two 2.00GHz Intel(R) Xeon(R) CPUs and a Nvidia Tesla P100-PCIE-16GB GPU. Consistent with

prior research[25, 33, 34, 36, 52], we employed the Area Under the Curve (AUC) as the primary evaluation metric, with Accuracy (ACC) serving as the secondary metric. We repeated the experiment five times and reported the average performance[2, 35].

## 4.4 Experimental results

In this section, we compared ReKT with other baseline methods.

*4.4.1 Concept-based KT Performance.* We modify ReKT by removing question from its input, changing from $(q_t, c_t, r_t)$ to $(c_t, r_t)$. Here, $E_t = C_{c_t}$. As questions are absent, question knowledge state isn't traced in this setup. We call this adapted version ReKT-concept. We compare its performance with the concept-based KT baseline, as shown in Table 1. We can found: (1) Compared with other concept-based KT baseline models, ReKT-concept consistently achieves the best (in most cases) or nearly the best performance across all datasets. This underscores ReKT's remarkable capability in effectively tracing the knowledge states of students; (2) Notably, the performance of DKT+ excels on certain datasets (e.g., ASSIST09 and Statics2011), attributed to its assumption of uniform recent responses among students. However, this assumption doesn't truly capture the students' knowledge states; rather, it observes a certain response pattern. While DKT+'s performance thrives when a dataset adheres to this pattern, its efficacy substantially declines when the dataset deviates from it (as seen in ASSIST17 and Eedi).

**Table 3: Performance comparison of ReKT ablation study. "Q" means question knowledge state, "C" means concept knowledge state, and "D" means domain knowledge state. Best results in bold.**

| Q | C | D | ASSIST09 | | ASSIST12 | | ASSIST17 | | Statics2011 | | EdNet | | Eedi | |
|---|---|---|---|---|---|---|---|---|---|---|---|---|---|---|
| | | | AUC | ACC | AUC | ACC | AUC | ACC | AUC | ACC | AUC | ACC | AUC | ACC |
| √ | | | 0.6801 | 0.6745 | 0.7056 | 0.7208 | 0.7251 | 0.6641 | 0.7915 | 0.8120 | 0.7423 | 0.7275 | 0.6608 | 0.6584 |
| | √ | | 0.7694 | 0.7285 | 0.7691 | 0.7504 | 0.7455 | 0.6834 | 0.8603 | 0.8325 | 0.7390 | 0.7272 | 0.7467 | 0.7053 |
| | | √ | 0.7823 | 0.7388 | 0.7771 | 0.7570 | 0.7740 | 0.7045 | 0.8950 | 0.8561 | 0.7647 | 0.7385 | 0.7945 | 0.7385 |
| √ | √ | | 0.7701 | 0.7274 | 0.7691 | 0.7502 | 0.7462 | 0.6840 | 0.8592 | 0.8335 | 0.7459 | 0.7296 | 0.7449 | 0.7035 |
| √ | | √ | 0.7815 | 0.7380 | 0.7780 | 0.7582 | 0.7803 | 0.7094 | 0.8955 | 0.8562 | 0.7745 | 0.7444 | 0.7946 | 0.7382 |
| | √ | √ | 0.7915 | 0.7440 | 0.7846 | 0.7607 | 0.7793 | 0.7093 | 0.8962 | 0.8566 | 0.7689 | 0.7426 | 0.7965 | 0.7394 |
| √ | √ | √ | **0.7917** | **0.7449** | **0.7852** | **0.7609** | **0.7814** | **0.7102** | **0.8967** | **0.8568** | **0.7752** | **0.7447** | **0.7971** | **0.7397** |

**Table 4: Performance comparison of ReKT-concept ablation study. "C" means concept knowledge state, and "D" means domain knowledge state. Best results in bold.**

| C | D | ASSIST09 | | ASSIST12 | | ASSIST15 | | ASSIST17 | | Statics2011 | | EdNet | | Eedi | |
|---|---|---|---|---|---|---|---|---|---|---|---|---|---|---|
| | | AUC | ACC | AUC | ACC | AUC | ACC | AUC | ACC | AUC | ACC | AUC | ACC | AUC | ACC |
| √ | | 0.7493 | 0.7157 | 0.7159 | 0.7267 | 0.7395 | 0.7570 | 0.6771 | 0.6392 | 0.8044 | 0.8084 | 0.6722 | 0.7060 | 0.7134 | 0.6887 |
| | √ | 0.7680 | 0.7321 | 0.7295 | 0.7354 | 0.7271 | 0.7544 | 0.7156 | 0.6653 | 0.8513 | 0.8293 | 0.7041 | 0.7132 | 0.7661 | 0.7187 |
| √ | √ | **0.7737** | **0.7340** | **0.7359** | **0.7385** | **0.7531** | **0.7624** | **0.7237** | **0.6690** | **0.8546** | **0.8319** | **0.7096** | **0.7153** | **0.7693** | **0.7208** |

In contrast, ReKT-concept consistently performs well across all datasets, indicating its genuine ability to trace students' knowledge states; (3) The promising DKT+forgetting performance emphasizes the essential role of incorporating knowledge state forgetting mechanisms in KT models; (4) In most scenarios, AKT-concept outperforms SAINT-concept, implying that the sole application of the Transformer architecture yields marginal improvements for KT. The context-aware Transformer architecture introduced by AKT more effectively captures students' knowledge states; (5) The impressive performance of ATKT indicates that adversarial training enhances KT models' generalization capability.

*4.4.2 Question-based KT Performance.* Table 2 presents a performance comparison between ReKT and other question-based KT baseline models. Due to the unavailability of question data in ASSIST15, it is not applicable to question-based KT. We can draw the following conclusions from Table 2: (1) Compared to other question-based KT baseline models, ReKT demonstrates significantly superior performance. This highlights the excellence of ReKT, despite its inherent simplicity; (2) It is evident that AKT demonstrates powerful performance, but it relies on a specialized architecture: Context-Aware Transformer. However, this architecture is highly complex. In comparison, FRU is particularly simple; (3) The impressive performance of GIKT and CDKT underscores the advantageous role of effectively representing questions in enhancing the performance of KT models; (4) Overall sound performance of DIMKT suggests the utility of incorporating question difficulty information within the KT framework; (5) The consistent performance of QIKT indicates the effectiveness of modeling students' knowledge state around the questions; (6) Comparing ReKT with simpleKT, despite the simplicity of simpleKT in comparison, ReKT notably outperforms simpleKT and strikes a balance between efficiency and simplicity in its architecture, a characteristic lacking in simpleKT.

## 4.5 Ablation Study

In this section, we explore the influence of various knowledge states on ReKT. Specifically, in question, concept, and domain knowledge states, we select one or two of them as variants to contrast with ReKT for ablation experiments. The ablation experiments for ReKT are shown in Table 3, while the ablation experiments for the ReKT-concept are shown in Table 4. Please note that ReKT-concept, as it does not include questions, does not trace the question knowledge state. We can observe: (1) When tracing only one knowledge state, the performance of question, concept, and domain knowledge states shows an increasing trend. This is evident as the interaction history data they utilize also increases in the same order. The model evidently benefits from more data, thus leading to better performance with domain knowledge state; (2) Tracing two knowledge states, as opposed to one, results in a notable performance improvement. Furthermore, the performance of question + domain knowledge state is not always inferior to concept + domain knowledge state (as observed in ASSIST17 and EdNet). This indicates that across different datasets, each knowledge state plays a significant role; (3) Tracing all three of these knowledge states (ReKT) yields significant improvements in performance across all datasets compared to tracing only one or two knowledge states. This undoubtedly demonstrates the effectiveness of these three knowledge states and validates the effectiveness of the proposed multi-perspective modeling.

## 4.6 Core architecture performance for KT

In this section, we will explore the performance of different core architectures on KT. Specifically, we will compare FRU with four commonly used core architectures in KT: LSTM, GRU, Transformer, and AKT-Transformer [12], while only tracing domain knowledge state (as is the practice in most KT research). The results for question-based KT are shown in Table 5, and the results for concept-based KT can be found in Table 6. We can conclude the following: (1) In terms of an overall performance comparison, in question-based KT, FRU as the core architecture achieves the best performance in most cases. In concept-based KT, FRU also demonstrates commendable performance. This suggests that the proposed FRU architecture is highly suitable for KT; (2) When comparing the number of parameters and computing resources, FRU requires significantly fewer parameters and computing resources. FRU achieves excellent performance with

**Table 5: Performance of different core architectures for question-based KT. "AKT-Transformer" means context-aware Transformer architecture in AKT, which modifies the calculation of basic Transformer attention scores to consider the forgetting behavior of students based on contextual information. "Rank" reports the average rank across all datasets. Best results in bold.**

| Core Architecture | ASSIST09 AUC | ASSIST09 ACC | ASSIST12 AUC | ASSIST12 ACC | ASSIST17 AUC | ASSIST17 ACC | Statics2011 AUC | Statics2011 ACC | EdNet AUC | EdNet ACC | Eedi AUC | Eedi ACC | Rank | # params | FLOPs |
|---|---|---|---|---|---|---|---|---|---|---|---|---|---|---|---|
| LSTM | 0.7811 | 0.7384 | 0.7747 | 0.7566 | 0.7723 | 0.7033 | 0.8791 | 0.8451 | 0.7650 | 0.7382 | 0.7951 | 0.7377 | 2.5 | 165.121K | 2.643G |
| GRU | **0.7837** | 0.7381 | 0.7762 | 0.7558 | 0.7720 | 0.7035 | 0.8785 | 0.8461 | **0.7658** | 0.7383 | **0.7957** | 0.7368 | 2.0 | 132.097K | 2.115G |
| Transformer | 0.7635 | 0.7231 | 0.7648 | 0.7483 | 0.7594 | 0.6944 | 0.8638 | 0.8382 | 0.7612 | 0.7362 | 0.7808 | 0.7267 | 5.0 | 627.841K | 9.963G |
| AKT-Transformer | 0.7738 | 0.7336 | 0.7765 | 0.7567 | 0.7623 | 0.6935 | 0.8768 | 0.8454 | 0.7615 | 0.7363 | 0.7903 | 0.7344 | 3.7 | 627.841K | 9.963G |
| FRU | 0.7823 | **0.7388** | **0.7771** | **0.7570** | **0.7740** | **0.7045** | **0.8950** | **0.8561** | 0.7647 | **0.7385** | 0.7945 | **0.7385** | **1.8** | **98.817K** | **1.567G** |

**Table 6: Performance of different core architectures for concept-based KT. Refer to Table 5 for a detailed explanation.**

| Core Architecture | ASSIST09 AUC | ASSIST09 ACC | ASSIST12 AUC | ASSIST12 ACC | ASSIST15 AUC | ASSIST15 ACC | ASSIST17 AUC | ASSIST17 ACC | Statics2011 AUC | Statics2011 ACC | EdNet AUC | EdNet ACC | Eedi AUC | Eedi ACC | Rank | # params | FLOPs |
|---|---|---|---|---|---|---|---|---|---|---|---|---|---|---|---|---|---|
| LSTM | **0.7684** | 0.7297 | 0.7328 | 0.7367 | 0.7323 | 0.7536 | 0.7188 | 0.6673 | 0.8483 | 0.8275 | 0.7006 | 0.7129 | 0.7629 | 0.7182 | 2.4 | 165.121K | 2.643G |
| GRU | 0.7674 | 0.7314 | **0.7389** | **0.7397** | 0.7331 | 0.7568 | **0.7192** | **0.6686** | 0.8428 | 0.8258 | 0.7037 | **0.7136** | 0.7654 | 0.7180 | **2.0** | 132.097K | 2.115G |
| Transformer | 0.7487 | 0.7140 | 0.7289 | 0.7353 | **0.7365** | **0.7572** | 0.7013 | 0.6560 | 0.7598 | 0.7940 | 0.6974 | 0.7096 | 0.7589 | 0.7130 | 4.4 | 627.841K | 9.963G |
| AKT-Transformer | 0.7668 | 0.7280 | 0.7384 | 0.7390 | 0.7312 | 0.7557 | 0.7157 | 0.6637 | 0.8384 | 0.8204 | 0.6987 | 0.7111 | 0.7626 | 0.7166 | 3.6 | 627.841K | 9.963G |
| FRU | 0.7680 | **0.7321** | 0.7295 | 0.7354 | 0.7271 | 0.7544 | 0.7156 | 0.6653 | **0.8513** | **0.8293** | **0.7041** | 0.7132 | **0.7661** | **0.7187** | 2.6 | **98.817K** | **1.567G** |

**Table 7: Performance of different core architectures for ReKT. "AKT-Transformer" means context-aware Transformer architecture in AKT, which modifies the calculation of basic Transformer attention scores to consider the forgetting behavior of students based on contextual information. Model names are formed from core architecture initials. Best results in bold.**

| Model | Core Architecture | ASSIST09 AUC | ASSIST09 ACC | ASSIST12 AUC | ASSIST12 ACC | ASSIST17 AUC | ASSIST17 ACC | Statics2011 AUC | Statics2011 ACC | EdNet AUC | EdNet ACC | Eedi AUC | Eedi ACC | # params | FLOPs |
|---|---|---|---|---|---|---|---|---|---|---|---|---|---|---|---|
| ReKT-L | LSTM | 0.7874 | 0.7443 | 0.7794 | 0.7591 | 0.7766 | 0.7061 | 0.8825 | 0.8486 | 0.7722 | 0.7435 | 0.7968 | 0.7383 | 0.592M | 9.425G |
| ReKT-G | GRU | **0.7933** | **0.7465** | 0.7835 | 0.7607 | 0.7790 | 0.7073 | 0.8823 | 0.8493 | 0.7745 | 0.7441 | **0.7982** | **0.7404** | 0.461M | 7.332G |
| ReKT-T | Transformer | 0.7629 | 0.7238 | 0.7536 | 0.7420 | 0.7638 | 0.6958 | 0.8322 | 0.8254 | 0.7679 | 0.7387 | 0.7793 | 0.7205 | 1.454M | 23.078G |
| ReKT-A | AKT-Transformer | 0.7702 | 0.7318 | 0.7669 | 0.7510 | 0.7656 | 0.6965 | 0.8602 | 0.8381 | 0.7680 | 0.7409 | 0.7892 | 0.7338 | 1.454M | 23.078G |
| ReKT(ours) | FRU | 0.7917 | 0.7449 | **0.7852** | **0.7609** | **0.7814** | **0.7102** | **0.8967** | **0.8568** | **0.7752** | **0.7447** | 0.7971 | 0.7397 | **0.263M** | **4.175G** |

**Table 8: Performance of different core architectures for ReKT-concept. Refer to Table 7 for a detailed explanation.**

| Model | Core Architecture | ASSIST09 AUC | ASSIST09 ACC | ASSIST12 AUC | ASSIST12 ACC | ASSIST15 AUC | ASSIST15 ACC | ASSIST17 AUC | ASSIST17 ACC | Statics2011 AUC | Statics2011 ACC | EdNet AUC | EdNet ACC | Eedi AUC | Eedi ACC | # params | FLOPs |
|---|---|---|---|---|---|---|---|---|---|---|---|---|---|---|---|---|---|
| ReKT-concept-L | LSTM | 0.7685 | 0.7316 | 0.7352 | 0.7366 | 0.7493 | 0.7603 | 0.7196 | 0.6680 | 0.8363 | 0.8233 | 0.7080 | 0.7150 | 0.7679 | 0.7199 | 0.379M | 6.034G |
| ReKT-concept-G | GRU | 0.7724 | 0.7328 | 0.7343 | 0.7368 | 0.7511 | 0.7618 | 0.7212 | 0.6687 | 0.8330 | 0.8182 | 0.7084 | 0.7149 | **0.7704** | **0.7229** | 0.296M | 4.724G |
| ReKT-concept-T | Transformer | 0.7627 | 0.7271 | 0.7243 | 0.7323 | 0.7287 | 0.7553 | 0.7101 | 0.6574 | 0.7987 | 0.8100 | 0.6992 | 0.7125 | 0.7587 | 0.7144 | 1.239M | 19.662G |
| ReKT-concept-A | AKT-Transformer | 0.7689 | 0.7306 | 0.7339 | 0.7367 | 0.7316 | 0.7561 | 0.7169 | 0.6656 | 0.8335 | 0.8220 | 0.7014 | 0.7121 | 0.7653 | 0.7183 | 1.239M | 19.662G |
| ReKT-concept(ours) | FRU | **0.7737** | **0.7340** | **0.7359** | **0.7385** | **0.7531** | **0.7624** | **0.7237** | **0.6690** | **0.8546** | **0.8319** | **0.7096** | **0.7153** | 0.7693 | 0.7208 | **0.181M** | **2.871G** |

approximately 38% of the computing resources compared to other core architectures (the following sections will present the calculation process). This undoubtedly proves the lightweight nature of FRU and highlights its simplicity and effectiveness as the core architecture; (3) Considering the average rank, FRU performs the best in question-based KT and ranks relatively well in concept-based KT. This indicates that FRU maintains strong competitiveness throughout, despite it is very simple (details on the computing resources and ranking methodology are clarified in the following sections).

*4.6.1 Computing Resources.* We use the number of parameters and FLOPs as reference computing resources. Additionally, we compare the resource utilization of FRU with two widely used core structures in knowledge tracing: LSTM and Transformer (LSTM is more commonly used than GRU in recurrent neural networks for KT). Therefore, the computing resources required for FRU are approximately $((98.817/165.121+1.567/2.643)/2+(98.817/627.841+1.567/9.963))/2 \approx 38\%$ times that of the other architectures.

*4.6.2 Rank.* We use AUC as the ranking metric, taking the average ranking of each core architecture across all datasets.

## 4.7 Different Core Architectures for ReKT

In this section, we discuss the impact of different core architectures on ReKT. Specifically, we compare four commonly used core architectures in KT: LSTM, GRU, Transformer, and AKT-Transformer, and their performance is shown in Table 7. The same experiments on ReKT-concept are presented in Table 8. Please note the difference between this experiment and the previous one: this experiment, in the context of ReKT, traces students' knowledge states from multiple perspectives, while the previous experiment, like most research, only traced domain knowledge states. We can draw the following conclusions: (1) Their performance follows the order of FRU, GRU, LSTM, AKT-Transformer, and Transformer in descending order, clearly demonstrating the effectiveness of FRU. In addition, simpler methods appear to be more effective. This could be attributed to the introduction of multiple perspectives, which provides more comprehensive information for these simplified models, while excluding unnecessary information or noise; (2) It also indicates that LSTM modeling outperforms AKT-Transformer or Transformer, consistent with much research [21, 22] showing that architectures based

on LSTM tend to perform better than those based on Transformers. This may be because LSTM aligns better with the characteristics of knowledge states: continuity and dynamic updates. (3) AKT-Transformer consistently outperforms Transformer, highlighting the importance of considering student knowledge forgetting behavior. (4) When comparing parameters and FLOPs, ReKT achieves optimal performance with minimal resources, highlighting its simplicity and effectiveness.

## 4.8 The time and space complexity of FRU

In this section, we will conduct a thorough analysis of the time and space complexity of FRU, which will aid us in gaining a deeper understanding of FRU. FRU is shown in Figure 3. We denote the time step as $T$, and for simplicity, we set the hidden layer dimension to be consistent as $d$. For any time step $t \in [1, T]$, $Response_t \in \mathbb{R}^{1 \times d}$, $X_t \in \mathbb{R}^{1 \times d}$, $Z_t \in \mathbb{R}^{1 \times d}$, $I_\alpha \in \mathbb{R}^{1 \times d}$.

**Learnable parameters of FRU**: The parameters that FRU learn are only $W_1 \in \mathbb{R}^{2d \times d}$, $b_1 \in \mathbb{R}^{1 \times d}$, $W_2 \in \mathbb{R}^{2d \times d}$, $b_2 \in \mathbb{R}^{1 \times d}$, then the total number of learnable parameters of FRU is $2d * d + 1 * d + 2d * d + 1 * d = 4d^2 + 2d$.

**Time complexity of FRU**: The complexity of calculating $f_t$ is $O(1 * 2d * d + 1 * d)$, the complexity of calculating $Response_t$ is $O(1 * d)$, and the complexity of calculating the update value (that is, $Tanh([Response_t \oplus X_t]W_2 + b_2))$ is $O(1 * 2d * d + 1 * d)$, and the complexity of calculating $Z_t$ is $O(1 * d)$. Then the complexity of one time step is $O((2d^2 + d) + d + (2d^2 + d) + d)$, that is, $O(4d^2 + 4d)$. Then the time complexity of FRU is $O(4Td^2 + 4Td)$.

**Space complexity of FRU**: The space complexity of $Response_t$, $X_t$, $Z_t$, and $I_\alpha$ is all $O(1 * d)$. Considering the total time step, their combined space complexity is $O(T * 4 * 1 * d)$. In addition, the number of learnable parameters of FRU is $4d^2 + 2d$, so the space complexity of FRU is $O(4Td + 4d^2 + 2d)$.

## 4.9 Visual Analysis

One of the most interesting applications of knowledge tracing is probably tracing students' knowledge states. Once a student's knowledge state is accurately captured, teachers can provide targeted guidance to address their weaknesses in understanding. In this section, we randomly selected students' interaction sequences on the ASSIST17 dataset for 15 time steps to compare the tracing of students' knowledge states between DKT-Q and ReKT. DKT-Q is a variant of DKT where the input is changed from concepts to questions, as shown in Figure 4 and Figure 5. The y-axis represents specific questions, differentiated by different colored circles. The x-axis represents student interactions (time step, question and question related concept). If the student answered the current question correctly, the corresponding circle is displayed. If the student answered incorrectly, an additional white circle is added to indicate the difference. The value of each element in the matrix represents the student's mastery of a specific question at the corresponding time step. As shown in Figure 4, DKT-Q can hardly trace the mastery of question 785 by the student (indicated by minimal changes in the student's mastery of question 785). This is because there is only one interaction about it in the interaction sequence (at time step $t = 13$). However, considering that question 785 is related to concept 78, and the student has actually practiced numerous exercises

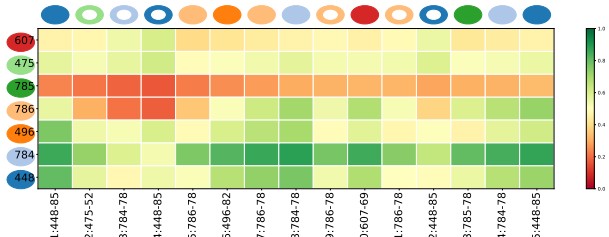

**Figure 4: A case study of DKT-Q for tracing knowledge state.**

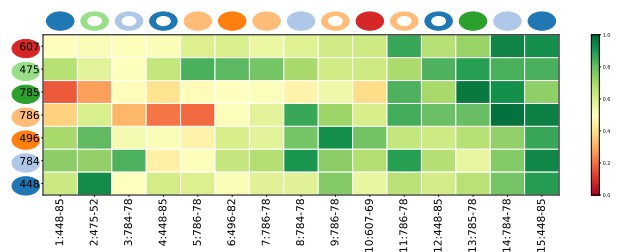

**Figure 5: A case study of ReKT for tracing knowledge state.**

related to concept 78 (at time steps $t = 3, 5, 7, 8, 9, 11$), it is intuitive to expect that the student's understanding of question 785 should improve. However, DKT-Q cannot capture this, whereas ReKT can trace the improvement in the student's mastery of question 785 as they continuously practice exercises related to concept 78 (as shown in Figure 5). In addition, during time steps $t = 5, 7, 9, 11$, the student practices question 786 continuously. Intuitively, regardless of whether the student answers correctly or incorrectly, the student should gain knowledge from it (improving the mastery of question 786). However, DKT-Q shows a decrease in the student's mastery of question 786 at time steps $t = 9, 11$, which goes against intuition. On the other hand, ReKT can continuously traces the improvement in the student's mastery of question 786. This shows the advantage of ReKT in tracing students' knowledge state. Through the visualization results, teachers can clearly understand the student's mastery of certain questions and provide targeted exercises.

## 5 CONCLUSION

In this paper, we revisit knowledge tracing and propose a simple and powerful model, ReKT. Firstly, inspired by the decision-making process of human teachers, we model students' knowledge states from three distinct perspectives: questions, concepts, and domains. Secondly, drawing inspiration from human cognitive development models, we design a lightweight FRU architecture as the core framework for KT tasks, comprising only two linear regression units. When compared to 22 state-of-the-art KT models on 7 publicly available datasets, the results indicate that ReKT achieves optimal performance in most cases, whether in question-based KT or concept-based KT. This underscores that, even without relying on complex models or cutting-edge technology, by delving deeper into the characteristics of KT tasks, one can design models that are both simple and powerful. We believe that ReKT has the potential to offer a wealth of new inspiration and insights for future KT research.

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
