# OpenReview forum: "Revisiting Knowledge Tracing: A Simple and Powerful Model"
_acmmm.org/ACMMM/2024/Conference — MM2024 Oral_

### Official Review · Reviewer_3xVL · 2024-05-24

**Rating:** 4
**Confidence:** 4

**Summary:**

This paper proposes a Knowledge Tracing (KT) model named ReKT, which aims to emphasize the representation of knowledge state and the simplicity and power of the core architecture.  First, in line with the decision-making process of human teacher, ReKT models knowledge state from question, concepts and domains and utilizes three types of subsequences. Moreover, the author designs a lightweight FRU module for ReKT. Extensive experiments on 7 datasets under different settings demonstrates the effectiveness of ReKT.

**Strengths:**

1. The paper is well-written and well-organized with good textual expression, mathematical notation, and clear figures.
2. The proposed method is highly acceptable and demonstrates some innovation. Considering students' knowledge state from these three angles aligns closely with educational practice. The proposed FRU, tailored for ReKT, effectively integrates the time interval of items within the subsequence, which complements the previous method.
3. The author conducts extensive experiments across 7 datasets, comparing against 22 baseline models under both concept-based and question-based settings to analyze RKT. The experimental results are good.

**Limitations:**

1. The motivation is not strong enough. Although I find the model framework proposed by the author intriguing, the motivation and challenges described in the introduction fails to engage me. It doesn't clearly articulate why simplifying the KT model is necessary and what problems complexity might pose for KT.
2. The article's core objective is to create a KT model that is both simple and powerful. However, the main experimental results only show the model's accuracy performance without including any metric to measure simplicity, such as parameter quantity in Table 5. I think this omission is somewhat inappropriate.
3.  Figure 4 and 5 need improvement in terms of aesthetics, as the circles in the graphs appear flattened. And some minor formatting issues need to be adjusted, such as removing spaces before citations.

**Suitability:**

2

---

### Official Review · Reviewer_mGeJ · 2024-05-26

**Rating:** 5
**Confidence:** 4

**Summary:**

This paper revisits knowledge tracing models and introduces the ReKT model. Drawing inspiration from the decision-making process of human teachers, the ReKT model traces knowledge states from three distinct perspectives. Additionally, inspired by constructivism, the innovative FRU architecture, tailored for knowledge tracing, is presented. Extensive experiments validate the effectiveness of this approach.

**Strengths:**

1.Online education or intelligent education is an important application scenario of multimedia technology, and the estimation of learners' cognitive states is a crucial issue in the application of intelligent education. This paper inspired by the decision-making process of human teachers, represents students' knowledge states from multiple perspectives, offering an intriguing research angle.

2.The ReKT model proposed in this paper, along with the FRU architecture, has been experimentally validated on multiple datasets and compared with numerous methods, demonstrating its effectiveness convincingly. The experimental data was comprehensive and highly credible. Extensive comparisons were conducted with 22 state-of-the-art KT models on 7 publicly available datasets.

3.This paper is concise and easy to read, with well-organized writing structure, overall presenting a high-quality work.

**Limitations:**

1.The FRU is interesting, but it can be further improved by considering learners' learning conditions such as learning rates to enhance its completeness. The author could explore these aspects further.

2.In Question-based Knowledge Tracing, further optimization of question representation should be considered, as this often has a positive impact on the model.

3. In the realm of sequence modeling, both RNNs and LSTMs have been widely acknowledged for their simplicity and effectiveness. However, it is imperative to consider the advantages that Fully Recurrent Unit (FRU) holds over these conventional models, particularly in the context of knowledge tracking. Please give more details.

Questions：

1.Some knowledge states, such as questions, are based on limited data, making it challenge to be well represented. How does the author consider meeting this challenge?

2.What are the limitations of the ReKT model?

**Suitability:**

3

---

### Official Review · Reviewer_4nLy · 2024-05-28

**Rating:** 4
**Confidence:** 4

**Summary:**

The paper proposes a simple yet effective model for knowledge tracing  (termed ReKT), which starts with a newly different research perspective for KT, i.e., considering multiple levels of the student's ability representations (including the question, concept, and domain) instead of a single level.  Besides, a lightweight unit forget-response update (FRU) is devised to aggregate corresponding ability representations in three levels. The paper executed extensive experiments on seven datasets by comparing over ten KT models, and the experimental results show the proposed ReKT achieves promising performance while maintaining good efficiency and less model complexity.

**Strengths:**

1) The motivation of this paper is clearly introduced
2) The proposed ReKT model is  simple  regarding model architecture yet effective  regarding model performance as well as model interpretability .
3) The sufficient and comprehensive evaluations executed in the experiments.

**Limitations:**

1) The difference between GRU and FRU is not clarified clearly;
2) Some important KT approaches through neural architecture search are missing in the related work;
3) There are some writing issues in the manuscript affecting the paper qualiity: on line 316, the sentence lacks main statement; on lines 265, 269, a blank space should be added  before citations; on line 327, the sentence is written too colloquially; and so many others. The author may double-check their paper to revised these issues.
4) The relation between this paper and ACM MM may be highlighted.

**Suitability:**

2

---

### Official Review · Reviewer_6eo9 · 2024-06-01

**Rating:** 5
**Confidence:** 3

**Summary:**

Advances in multimedia technology have made multimedia learning crucial. Knowledge Tracing (KT) monitors knowledge acquisition and predicts student performance. While current KT research focuses on complex deep learning models, this paper addresses fundamental KT challenges: knowledge state representation and core architecture design. The proposed ReKT model, inspired by human teachers, models student knowledge from questions, concepts, and domains. Using a lightweight Forget-Response-Update (FRU) framework, ReKT not only outperforms 22 state-of-the-art KT models on seven datasets, but does so with only 38% of the computing resources of other architectures. This superior performance offers new insights for future KT research. The code is available in supplementary materials.

**Strengths:**

1. Exquisite and precise diagrams in Figure 1 and Figure 2 clearly explain the concept of this task and the proposed ReKT model.
2. Starting from the basic concepts of the task, structuring the Knowledge Tracing (KT) problem with new ideas without using the popular and influential models of today is an excellent approach in itself.
3. The main learnable parameters are within the Forget-Response-Update (FRU) framework. This paper introduces a new concept that combines the use of the FRU framework, significantly reducing inference time while improving performance.
4. A wide range of experiments were conducted to verify the effectiveness and efficiency of the proposed method and detailed framework, resulting in significant improvements across most datasets. The proposed Forget-Response-Update (FRU) was also compared with existing frameworks with similar functionalities, demonstrating notable enhancements in effectiveness and efficiency.

**Limitations:**

1. What is the unique proposition behind the creation of a new concept for the combination of old concepts? Could this hinder the model from establishing relationships between concepts? Or is it a case where the task does not necessitate such relationships?
2. In the ablation study of replacing the core architecture, Tables 5 and 6 demonstrate the performance of the model in question-based and concept-based scenarios. The Forget-Response-Update (FRU) proposed in this work performs relatively poorly in the concept-based setting. Are there any underlying reasons for this?

**Suitability:**

2

---

### Meta-Review · Area_Chair_vv12 · 2024-07-04

**Recommendation:** Accept (Oral)
**Confidence:** 4

**Metareview:**

The paper proposes a simple yet effective model for knowledge tracing (termed ReKT). The proposed ReKT model, inspired by the decision-making processes of human teachers, provides a novel method of representing student knowledge from multiple perspectives, supported by the lightweight Forget-Response-Update (FRU) framework. This framework notably enhances performance while significantly reducing computational requirements, as demonstrated through comprehensive experiments across seven datasets. The reviewers have acknowledged the strengths of the paper, including its clear motivation, innovative approach, and extensive evaluation, which collectively validate the effectiveness and efficiency of the proposed model, while some minor limitations and suggestions for improvement have also been noted.